# *Adansonia digitata* L. (Baobab Fruit) Effect on Postprandial Glycemia in Healthy Adults: A Randomized Controlled Trial

**DOI:** 10.3390/nu14020398

**Published:** 2022-01-17

**Authors:** Keyla Rita, Maria Alexandra Bernardo, Maria Leonor Silva, José Brito, Maria Fernanda Mesquita, Ana Maria Pintão, Margarida Moncada

**Affiliations:** Centro de Investigação Interdisciplinar Egas Moniz, Instituto Universitário Egas Moniz, Campus Universitário, Quinta da Granja, Monte de Caparica, 2829-511 Caparica, Portugal; keyla.rita@gmail.com (K.R.); abernardo@egasmoniz.edu.pt (M.A.B.); lsilva@egasmoniz.edu.pt (M.L.S.); britojaa@hotmail.com (J.B.); fmesquita@egasmoniz.edu.pt (M.F.M.); apintao@egasmoniz.edu.pt (A.M.P.)

**Keywords:** *Adansonia digitata*, baobab fruit, postprandial glycemia, antioxidant, polyphenols, proanthocyanins

## Abstract

Baobab fruits have been traditionally used in Africa due to their therapeutic properties attributed to their high polyphenol content. The aim of the study was to investigate the effect of baobab fruit on postprandial glycaemia in healthy adults and to measure its bioactive compounds and antioxidant activity. The study (NCT05140629) was conducted on 31 healthy subjects. The participants were randomly allocated in the control group (oral glucose tolerance test (OGTT); *n* = 16) and in the intervention group (OGTT, followed by administration of 250 mL baobab aqueous extract (BAE); *n* = 15). Total phenols, proanthocyanins, hydrolyzable tannins, and antioxidant activity (FRAP, DPPH, ABTS, and inhibition of O2^•^^−^ and NO^•^ methods) were quantified. Repeated measures ANOVA of mixed type and independent samples t-test were used. Glycemia incremental area under the curve (*p* = 0.012) and glucose maximum concentration (*p* = 0.029) was significantly lower in the intervention group compared to the control group. The BAE revealed high total contents of phenols, proanthocyanins, and hydrolyzable tannins, as well as a strong capacity to scavenge superoxide anions and nitric oxide radicals and a high antioxidant activity. In conclusion, this study encourages the use of this food component as a promising source of natural antioxidants and a hypoglycemic agent under glucose load acute conditions.

## 1. Introduction

Traditional plants have been used as potential preventive and therapeutic agents in blood glucose management for hundreds of years. A large number of medicinal plants have been demonstrated to be effective in diabetes mellitus (DM) treatment, showing a hypoglycemic effect [1]. The daily use of these plants is important since DM prevalence is increasing worldwide, causing serious complications which reduce quality of life and increase mortality [2].

*Adansonia digitata* L. (baobab) is an emblematic and cultural tree widely distributed throughout Africa and India and has increasing interest due to its medicinal uses [3,4]. This tree, belonging to the *Malvaceae* family, is native to the African continent [4] and is found mainly in low altitude regions, including the savannas of sub-Saharan Africa [5].

Baobab fruit has revealed promising results in controlling glycemia levels, although few studies have been published in the literature. Nevertheless, some promising results have been published, such as the aqueous extract of baobab fruit powder pulp significantly reducing (*p <* 0.05) blood glucose area under the curve (AUC) at 120 and 180 min in humans [6]; or the significant (*p <* 0.05) decrease of insulin response AUC, after ingestion of white bread enriched with polyphenol extract from baobab in healthy participants, in spite of not significantly reducing postprandial glucose response [7]. In addition, in animal models, methanolic extracts of baobab fruit pulp have also showed a hypoglycemic effect [8]. However, more recently, Evang et al. (2021) revealed that the ingestion of baobab fruit pulp did not significantly improve the hemoglobin level in the intervention group compared with the control group [9].

Baobab fruit seems to also possess potential antioxidant properties, having been attributed to its bioactive compounds, such as polyphenols [10]. Although the scarce information about its phenolic compounds [6], the procyanidins, such as epicatechin, procyanidin A2, procyanidin B2, procyanidin B5, procyanidin C1 [11], and quercetin, were found in baobab extracts [8]. This fruit also contains tannins, phytic acid, trypsin inhibitors, BAPA (N-benzoyl-DL-arginine-P-nitroanilide), trypsin type III [12], oxalates, and hydrocyanic acid, as well as has a high soluble fiber content [12,13].

Thus, we hypothesized that the ingestion of baobab aqueous extract could improve glycemia response in adult subjects, due to its content in polyphenol compounds, including proanthocyanin and hydrolyzable tannins. In order to test the hypothesis, a randomized controlled clinical trial was carried out in healthy adults ingesting *Adansonia digitata* L. fruit aqueous extract after an oral glucose tolerance test (OGTT), in which we studied the: (1) glycemic effect, (2) bioactive compounds, and (3) antioxidant activity.

## 2. Materials and Methods

### 2.1. Ethical Consideration

This study was approved by Egas Moniz Cooperativa de Ensino Superior Ethics Committee (protocol code 518) and was carried out in accordance with the Declaration of Helsinki (Declaration of 1975, revised in 2000). An informed consent was given to all eligible participants, after oral and written information about the study. This study was registered at Clinicaltrials.gov (NCT05140629).

### 2.2. Study Design and Participants

A randomized controlled clinical trial, blind to the researcher who performed the statistical analysis, was conducted with 31 subjects, recruited at the Campus Universitário Egas Moniz, in Monte de Caparica, Portugal. The eligibility criteria included male or female adults aged 18–40 years old, not currently undergoing lactation or pregnancy, and the ability to read and sign the informed consent. Healthy subjects, with a fasting blood glucose <126 mg/dL, were included. Subjects that were less than 8 or more than 10 h fasting, had symptoms and history of gastrointestinal, hepatic, and cardiovascular diseases, baobab intolerance or allergy, drug and/or supplement consumption capable of influencing plasma glucose, and had ingested baobab, water, coffee, or alcohol intake and smoked tobacco within 8 h before the intervention were excluded.

After eligibility criteria were met and inform consent signed, participants were subjected to the inclusion and exclusion criteria and subsequently randomly allocated into the control or intervention groups. Participants were sequentially assigned to the intervention or control group as they were recruited. A codification was attributed to each participant, in order to maintained anonymity and ensure confidentiality of data.

After a period of 8–10 h fasting, the control group was given OGTT, and the intervention group was given OGTT, followed by baobab aqueous extract.

The flow diagram enrolment, allocation, follow-up, and analysis of this study’s participants are shown in Figure 1.

### 2.3. Baobab Aqueous Extract Preparation

The baobab fruit was bought in the market of Benfica, located in the city of Luanda (Angola), and brought to Lisbon (Portugal), duly packed in a plastic bag and stored under standard environmental conditions (21–23 °C, 50–60% humidity) until needed.

For the aqueous extraction of the fruit 40 ± 1 g of pulp, seeds and red filaments were weighed, and the fruit was boiled in 300 mL of water for 5 min. After slowly cooling until room temperature, the extract was then placed in a refrigerator in a container, where it remained for 8 h at an average temperature of 10 °C. The BAE was sieved with the aid of a net sieve, separating it from the seeds and filaments, and was subsequently subjected to the clinical trial and the antioxidant assays. For the antioxidant assays, after filtration, the extract was passed through the blender (Kenwood Multipro Compact Food Processor FDP302SI) and filtered again to obtain homogeneous samples, avoiding the formation of precipitates during the analysis. The final concentration of the aqueous extract obtained, for both clinical trial and chemical analysis, was 0.1333 g *Adansonia digitata* L. (AD)*/mL* extract fresh weight (FW).

### 2.4. Interventions

After overnight fasting, blood glucose level was assessed through a capillary drop blood, immediately before an Oral Glucose Tolerance Test (t0). The control group participants ingested glucose solution alone (OGTT), prepared with 75 g of anhydrous oral glucose, as prescribed by the ADA [14], dissolved in 200 mL of water. The intervention group ingested glucose solution, followed by 250 mL of baobab aqueous extract (33.33 g FW). Blood glucose level was also measured at 30 (t30), 60 (t60), 90 (t90), and 120 (t120) min immediately following the intervention, for each participant, in the control and intervention groups. Glucometer equipment (Lisbon, Portugal), strips for glucose meters (One Touch Select Plus) (Lisbon, Portugal), and sterilized lancets (Sarstedt normal 21G) (Lisbon, Portugal) were used to measure blood glucose concentrations, taking due care of safety and asepsis.

Based on the glycemia values, the blood glucose incremental area under the curve (AUCi) of each participant was defined using GraphPad Prism (version 5.0) (San Diego, USA). Maximum concentrations (Cmax) and variations of maximum concentrations (ΔCmax) were determined by comparing them with their respective baseline glycemia levels values.

### 2.5. Anthropometric Parameters Assessment

The anthropometric parameters were collected, namely weight, height, and body mass index (BMI). BMI was calculated as weight (Kg) divided by height (m^2^) squared (Kg/m^2^). Body weight was estimated by bio-impedance, through the Inbody^®^ scale, model 230 (Seoul, Korea).

### 2.6. Dietary Ingestion Assessment

Participants of the study completed a 24-h food recall questionnaire, which were carefully instructed by an investigator to identify all food consumed. The amount of each food ingested was estimated with help from pictures. A book with pictures of meals in different sizes was used. The investigator reviewed the questionnaire together with the participant.

Food intake of each participant was analyzed using The Food Processor SQL (version 11.3.285) software (Oak Brook, IL, USA), obtaining total energy, carbohydrates, proteins, and lipids mean intake.

### 2.7. Chemical Analysis

Folin–Ciocalteu and gallic acid-1-hydrate (C6H2(OH)3COOH.H2O) were from PanReac (Castellar del Vallès, Barcelona); tannic acid, TPTZ 2,4,6-tri (2-pyridyl)-s-triazine, Trolox (6-hydroxy-2,5,7,8-tetramethylchroman-2-carboxylic acid), DPPH (2,2-diphenyl-1-picrylhydrazyl), ABTS (2,2′-azino-bis (3-ethylbenzo thiazoline-6-sulfonic acid), potassium persulfate (K2S2O8), NADH (nicotine-adenine dinucleotide reduced), NBT (nitro blue tetrazolium) ≥97%, and PMS (phenazine methosulfate) were from Sigma-Aldrich (Buenos Aires, Argentina); 1-butanol, NED (1- naphthyl ethylenediamine dihydrochloride), and sulfanilic acid were from Merck (Darmstadt, Germany); procyanidin A2 was from Extrasynthese (Genay, França); potassium iodate was from VWR Analar Normapur; and sodium nitroprusside was from Riedel-de Haën (Charlotte, NC, USA). All reagents were pro analysis grade. All absorbance measurements were performed in a Perkin-Elmer Lambda 25 (Watertown, MA, USA). The reagents were weighed on an analytical balance (Sartorius ± 0.0001 g).

### 2.8. Total Phenol, Proanthocyanidins, and Hydrolysable Tannins Content Assessment

Total phenolic concentration was determined according to the Folin–Ciocalteu method, employing gallic acid as the standard [15]. Results were expressed as mg of gallic acid equivalents/L (GAE/L). For this analysis, 125 μL of BAE, previously diluted (×15) in water, and 125 μL of ethanol was added to 2.5 mL of Folin–Ciocalteu reagent solution (1:10 diluted in H_2_O) and 2 mL of aqueous sodium carbonate (Na_2_CO_3_) 1M. After 15 min, the absorbance was measured at 765 nm.

The content of proanthocyanins was determined according to Gu et al.’s (2002) method, with modifications, which is based on acidic hydrolysis of proanthocyanidins polymers producing reddish pigments in hot 1-butanol/HCl solution [16]. Then, 200 μL of diluted BAE (×2) was added to 200 μL of methanol and 2600 μL of HCl/1-butanol 10%(*v*/*v*) solution. The test tubes were shaken and incubated, at 100 °C for 50 min. The absorbance was measured at 550 nm. Results are expressed as mg equivalents of procyanidin A2/L (EPA2/L).

The method of Willis and Allen was adapted for the determination of hydrolyzable tannins of the BAE [17]. A volume of 1 mL of extract was added to 5 mL of 2.5% potassium iodate (KIO_3_). The mixture was then stirred and returned to the water bath at 25 °C for 20 min. An absorbance reading was performed by a spectrophotometer (Sturbridge, MA, USA) at 550 nm, and the results were expressed as mg of tannic acid equivalents/L (TAE/L).

### 2.9. Antioxidant Assays

The antioxidant effect (reducing ability) was evaluated by monitoring the formation of an intense blue color from the Fe^2+^ TPTZ complex, according to the Ferric Reducing Antioxidant Power (FRAP) assay [18]. A volume of 2850 µL (25 mL 300 mM acetate buffer pH = 3.6 solution plus 2.5 mL 10 mM TPTZ solution in HCl 40 mM + 2.5 mL 20 mM FeCl_3_.6H_2_O solution) was added to 150 µL of diluted BAE. The tubes were kept in the dark for 30 min (7 assays at different concentrations). The absorbance was determined at 593 nm, and the results were expressed in mg of Trolox equivalents/L (TE/L).

The ABTS method was based on the capacity of a sample to inhibit the ABTS radical (ABTS^+^) compared with a reference antioxidant standard (Trolox). The ABTS^+^ radical was generated by chemical reaction with potassium persulfate (K_2_S_2_O_8_). Thus, 25 mL of ABTS (7 mM) was added to 440 µL of K_2_S_2_O_8_ (140 mM) and allowed to stand in darkness at room temperature for 12–16 h. The solution was prepared by taking a volume of the previous solution and diluting it in ethanol until its absorbance at λ = 734 nm was 0.70 [19]. The tubes were kept in the dark for 30 min. The results expressed in mg of Trolox equivalents/L (TE/L).

The DPPH method was determined by the 2,2-diphenyl-1-picrylhydrazyl radical scavenger [18]. A volume of 150 μL BAE was added to 2850 μL of DPPH solution previously prepared in methanol with a λ = 515 nm of 1.1. The solutions were kept for 24 h in the absence of light. The absorbance was determined at 515 nm, and the results expressed in mg of Trolox equivalents/L (TE/L).

### 2.10. Inhibition Capacity of O_2_^•^^−^ Anion and NO Radical

The superoxide anion is generated by oxidation of NADH by reacting with PMS and oxygen, causing the reduction of NBT [20,21]. A volume of 500 μL of BAE with different concentrations were added to 2 mL of NADH (189 μM) and NBT (120 μM) in 40 mM Tris-HCl buffer pH 8. The reaction was started after the addition of 0.5 mL of PMS (60 μM), and, after 5 min incubation at room temperature, the absorbance was measured at 560 nm.

The inhibition NO radical test was based on the method of Nikkhah and co-workers (2008) [22]. A volume of 250 μL of BAE with different concentrations was added to 1 mL of sodium nitroprusside and 250 μL of phosphate buffered saline (PBS). The mixture was kept for 150 min at 25 °C. Then, 0.5 mL of this mixture was added to 1 mL of sulfanilic acid (0.33% in 20% glacial acetic acid) and kept for 5 min at room temperature. Then, 1 mL of NED (0.1% *w*/*v*) was added, and this mixture was kept for 30 min at 25 °C. At the end of the reaction, a pink chromophore was formed. The absorbance was measured at 540 nm.

The percentage of inhibition of O2^•^^−^ and NO radicals was determined using Equation (1), and the results were expressed as mg of gallic acid equivalents (GAE)/L.
(1)$$\%I=\frac{A_{control}-A_{sample}}{A_{control}}\times 100$$

### 2.11. Statistical Analysis

Statistical analysis was performed using Excel^®^ (Washington, DC, USA) and SPSS^®^ (Statistical Package for Social Sciences) (New York, NY, USA), version 27.0, software, for Mac. Data are presented as mean ± SD and SEM. Repeated measurement ANOVA of mixed type was used to assess the difference between the 2 groups for postprandial blood glucose at different times. The independent samples *t*-test was used to assess the difference between the 2 groups for anthropometric parameters, total energy intake, carbohydrates, protein, and lipids, and Cmax, ΔCmax, and AUCi values. All statistical tests were performed at the 5% level of significance.

## 3. Results

### 3.1. Participants of the Study

A total of 31 subjects were randomly allocated to the study, with the control group constituting 16 (9 female, 7 male), and the intervention group 15 (12 female, 3 male). The general characteristics of the study participants are shown in Table 1. At baseline, no significant differences (*p >* 0.05) between the control and intervention groups were found regarding age, anthropometric parameters, and dietary intake.

### 3.2. Postprandial Glycemia

Postprandial blood glucose response for the intervention and control groups are shown in Table 2. The capillary blood glucose concentration did not significantly differ between groups at 0 min (baseline) (*p* = 0.115). The analysis of independent factors of repeated measures (ANOVA) revealed that there was no interaction between the independent and repeated measures factors (*p* = 0.092); thus, it is not possible to infer any differences in PBG in different moments. Nevertheless, baobab extract ingestion revealed a slightly decreased of capillary blood glucose level at different moments (Figure 2).

There was also no significant difference in the ΔCmax variation (*p* = 0.054) between the 2 groups. However, Cmax and AUCi mean values were significantly lower in the intervention group (*p* = 0.012 and *p* = 0.029, respectively), suggesting a promising effect of baobab ingestion on blood glucose control in healthy subjects (Table 3).

### 3.3. Compounds Quantification and Antioxidant Capacity of BAE

Baobab aqueous extract (BAE) chemical analysis revealed high total phenolic content and considerable proanthocyanidins and hydrolyzable tannins contents. Moreover, BAE showed high antioxidant activity by FRAP, DPPH, and ABTS methods, as well as a high inhibition capacity of anion O_2_^•^^−^ and NO^•^ radical (IC_50_) (Table 4).

## 4. Discussion

Baobab fruit appears to be effective on postprandial glycemia response [6], although scarce randomized clinical trials have been published. In the present study, the ingestion of aqueous baobab extract (0.1333 g *Adansonia digitata* L./mL extract fresh weight) significantly reduced postprandial glycemia incremental area under the curve (AUCi) at 120 min (*p* = 0.012) and capillary blood glucose maximum concentration (*p* = 0.029), although no effect on variation of glucose maximum concentration (*p* > 0.05) was observed. In this context, this study supports the hypothesis that the ingestion of baobab aqueous extract may improve postprandial glycemia in healthy adults. These results are in according to Coe et al.’s (2013) study [6], in which the glycemic response AUC significantly reduced at 60, 120, and 180 min after ingestion of baobab extract. However, Coe et al. (2016) showed that the ingestion of white bread enriched with polyphenols extracts has no effect on postprandial glycemic response [7]. The results of the study should be considered with some caution as heterogeneity regarding male/female ratio between groups was not observed, due to random allocation and the simple size.

However, the main parameters that influenced glucose control were identified, and its homogeneity between groups was verified. Moreover, at baseline, there were no significant differences between groups on blood glucose levels (*p* > 0.05).

Indeed, it is well known that meal composition, particularly carbohydrate content, can influence glycemic response [23], as well as other nutritional factors, such as protein. According to the study by Karamanlis et al., protein intake significantly reduces postprandial glycemia through stimulation of glucagon-like peptide 1 (GLP-1) [24]. However, since baobab fruit has a residual protein content (3.2%) [12], this should not be the main factor in altering the glucose response.

Moreover, the beneficial effect on glycemic response can be attributed to the bioactive compounds identified in baobab extract [6,9,11]. Baobab fruit has been shown to be rich in polyphenols, such as epicatechin and procyanidins [11], which have gained interest due to their potential antihyperglycemic effect. Indeed, vitro and in vivo studies have reported that epicatechin and procyanidins B2 and C1 have been shown to promote insulin secretion by increasing the level of GLP1 [25,26], which contributes to the improvement of blood glucose response. Increased secretion of GLP-1 has also been reported after ingestion of polyphenol extracts when isolated from cocoa [27] and coffee [28]; in the latter case, the study reported that, in mouse cell lines (C57BL/6J), this increase is realized through a cAMP-dependent pathway.

Since the Angola baobab fruit extract analyzed in the present study has a high polyphenolic content, the authors suggest that these compounds may contribute to the increase of GLP-1 and, consequently, to insulin secretion, which may explain the hypoglycemic effect. However, as we did not assess the effect of baobab fruit extract on GLP-1 and insulin secretion, this hypothesis underlying the hypoglycemic effect should be addressed in further studies.

In addition, polyphenols in some fruits (e.g., apple and strawberry) appear to exert an effect on controlling glucose absorption. According to Manzano and Williamson (2010), there is a substantial inhibition of glucose transport by the glucose transporters SGLT1 and GLUT2 from the intestinal lumen to the cells [29]. Future studies should also investigate the effect of polyphenols from baobab fruit extract on enterocyte glucose transporters as it could be a possible way to explain the hypoglycemic effect found in the present study.

Moreover, according to Magaia (2013), baobab fruit has a high fiber content [30], which may also contribute to reduced glucose level due to inhibition of glucose absorption [31].

This study also showed that baobab extract possesses a strong antioxidant activity and reactive oxygen species inhibitory capacity. Considering that total phenols found in the present baobab extract were 702.39 ± 11.8 mg GAE/100 g, and that the major polyphenolic compounds in baobab are procyanidins (336.33 ± 10.85 mg EPA2/100 g) and hydrolyzable tannins (237.63 ± 4.71 mg TAE/100 g), we can suggest that the polyphenolic compounds found in this extract, particularly procyanidins and tannins, could be responsible for its antioxidant activity. According to Dudonné et al. (2009), significant correlations were found between DPPH, ABTS, and FRAP assays and total phenolic content. Furthermore, the authors indicate the existence of a relationship between polyphenolic compounds concentration of plants extract and its free radical scavenging and ferric reducing capacities [32].

Procyanidin B2 can also exert a beneficial effect in oxidative stress, preventing advanced glycation-end product formation in pancreatic β -cells, which occur by reactive oxygen species during hyperglycemia [33]. Under hyperglycemic conditions, the oxidative stress is generated by the peroxidation and pro-oxidation processes [34]. As a result of the advanced glycation of proteins and lipids, the production of reactive oxygen species (ROS) and cellular mitochondrial dysfunction occurs [35]. In fact, Bräunlich and co-workers (2013) reported an inhibitory activity of procyanidins B2, B5, and C1 against ROS [34].

In addition, the baobab extract showed an inhibition capacity of O_2_^•^^−^ anion and NO radicals, which was also considered to be a relevant result. The oxidative stress status that occurs during oscillations of blood glucose levels [35] involves the formation of free radicals, which results from the inactivation reaction of the superoxide anion and nitric oxide [36]. During the state of postprandial hyperglycemia, production of endothelial dysfunction may occur, which has been demonstrated in healthy subjects. According to Ceriello et al., oxidative stress may be the main factor for this occurrence [37]. Therefore, the results support our hypothesis that baobab aqueous extract possess an antioxidant capacity through the inhibition of O2^•^^−^ anion and NO^•^ radicals, which can lead to oxidative stress prevention.

Our study has limitations that should be included in further studies. Firstly, the authors did not evaluate plasma glucagon-like peptide (GLP-1) and insulin concentration, which are important to verify the effect of this extract on GLP-1 and insulin secretion. Secondly, in the present study, we did not obtain a homogeneous sample between groups regarding gender. Therefore, a higher number of participants should be included in order to assure a homogeneous distribution. Finally, this study design permits the analysis of this extract effect under acute condition (OGTT). Thus, additional studies should address the baobab fruit extract effect for longer periods and as part of a mixed-meal daily intake.

## 5. Conclusions

The present study revealed that *Adansonia digitata* L. (baobab) fruit extract ingestion by healthy adults significantly reduces glycemia incremental AUC, as well as revealed its possessing a considerably antioxidant activity and inhibition capacity of reactive oxygen species. Therefore, the present results encourage the use of this food component as a promising source of natural antioxidants and a hypoglycemic agent under glucose load acute conditions.

## Figures and Tables

**Figure 1 nutrients-14-00398-f001:**
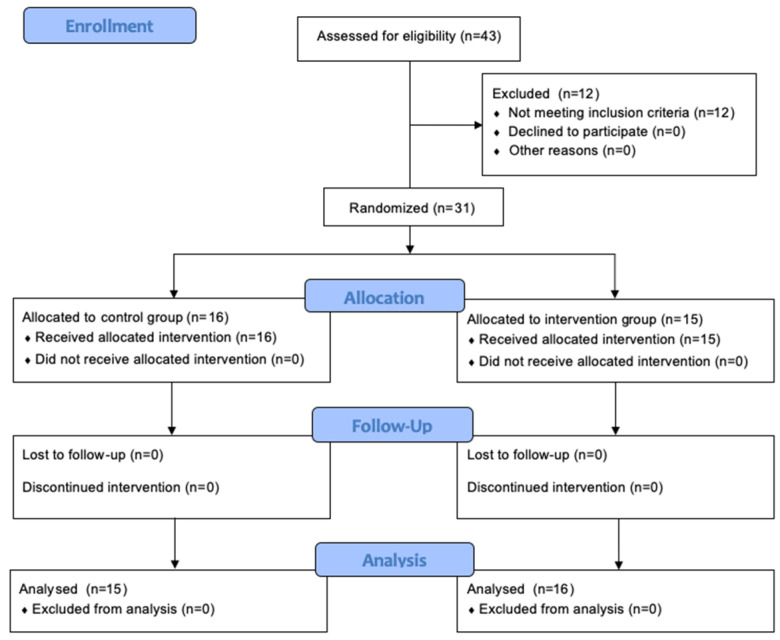
CONSORT flow diagram enrolment, allocation, follow-up, and analysis.

**Figure 2 nutrients-14-00398-f002:**
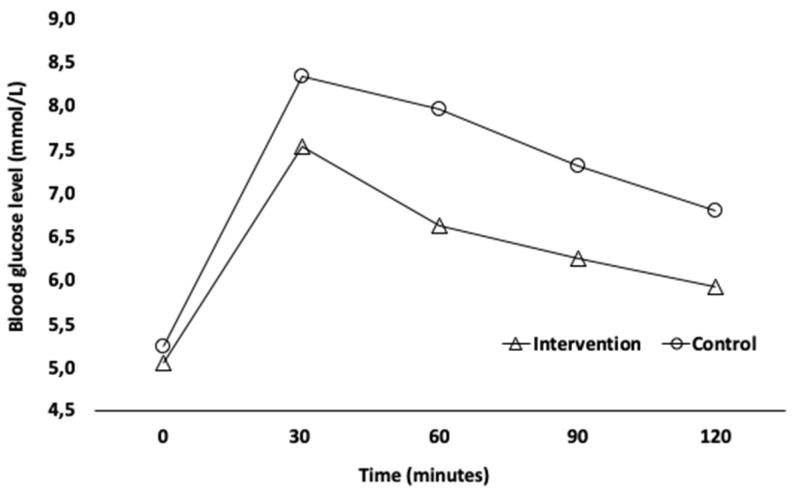
Mean time points of capillary blood glucose levels (mmol/L) in healthy adults.

**Table 1 nutrients-14-00398-t001:** Baseline characteristics of the study participants.

Variables	Control Group (*n* = 16) Mean (±SD)	Intervention Group (*n* = 15) Mean (±SD)	*p*-Value ^1^
Age (years)	25.25 (±7.29)	24.53 (±4.37)	0.744
Weight (m)	70.05 (±18.51)	65.36 (±9.67)	0.382
Height (m)	1.67 (±0.07)	1.65 (±0.08)	0.553
BMI (Kg/m^2^)	24.81 (±5.59)	23.82 (±3.33)	0.551
TEI at last meal before intervention (Kcal)	336.24 (±194.33)	595.05 (±578.01)	0.118
TEI * (Kcal)	1705.63 (±495.51)	1917.48 (±721.01)	0.353
Protein * (g)	85.21 (±33.40)	92.95 (±38.12)	0.554
Carbohydrate * (g)	208.40 (±88.14)	215.41 (±89.40)	0.828
Lipid * (g)	60.18 (±22.15)	78.34 (±35.56)	0.104

^1^*p*-Value was calculated by Student’s *t*-test. Abbreviations: BMI (body mass index); TEI (total energy intake); SD (standard deviation). * At the day before intervention.

**Table 2 nutrients-14-00398-t002:** Mean capillary blood glucose levels (mmol/L) obtained after an oral glucose tolerance test in the control group (*n* = 16), and after an oral glucose tolerance test plus baobab extract in the intervention group (*n* = 15), at different moments: before intervention (t0) and after 30 (t30), 60 (t60), 90 (t90), and 120 (t120) min after intervention.

Time	Control Group Mean (±SD) mmol/L	Max/Min Values mmol/L	Intervention Group Mean (±SD) mmol/L	Max/Min Values mmol/L
t0	5.25 (±0.39)	5.94/4.61	5.04 (±0.31)	5.94/4.61
t30	8.31 (±1.30)	10.71/5.49	7.54 (±1.00)	9.55/5.55
t60	7.93 (±1.56)	11.49/5.88	6.62 (±1.14)	8.33/4.38
t90	7.22 (±1.31)	9.93/4.77	6.24 (±0.61)	7.71/5.49
t120	6.75 (±1.19)	9.44/4.83	5.92 (±0.57)	6.99/4.83

*p*-Value was calculated by repeated measurement ANOVA of mixed type at different moments. Abbreviation: SD (standard deviation).

**Table 3 nutrients-14-00398-t003:** Capillary blood glucose level incremental area under the curve (AUCi), glucose maximum concentration (Cmax), and variation of glucose maximum concentration (ΔCmax) of participants.

Variables	Control Group (*n* = 16) Mean (±SD)	Intervention Group (*n* = 15) Mean (±SD)	*p*-Value ^1^
AUCi (0–120 min)	253.68 (±101.14)	172.44 (±61.92)	0.012
Cmax (mmol/L)	8.66 (±1.37)	7.71 (±0.85)	0.029
ΔCmax	3.41 (±1.17)	2.66 (±0.85)	0.054

^1^*p*-Value was calculated by independent samples *t*-test. Abbreviations: AUCi (incremental area under the curve); Cmax (maximum concentration); ΔCmax (variation of maximum concentration).

**Table 4 nutrients-14-00398-t004:** Total phenols, proanthocyanins, and hydrolyzable tannins content, anion and radical inhibition capacity, and antioxidant capacity of the BAE (0.133 g/mL FW), expressed as mean value (±SEM).

	Chemical Analysis	Mean (±SEM) /L	Mean (±SEM) /100 g
Compounds quantification	Total phenols (mg GAE, *n* = 7)	937 (±15.79)	702.39 (±11.85)
	Proanthocyanins (mg PAE2, *n* = 7)	448.43 (±14.46)	336.33 (±10.85)
	Hydrolysable tannins (mg TAE, *n* = 6)	316.84 (±6.28)	237.63 (±4.71)
Scavenging capacity	IC_50_ O_2_^•^^−^ anion (mg GAE, *n* = 7)	77.14 (±3.43)	57.86 (±2.57)
	IC_50_ NO^•^ radical (mg GAE, *n* = 5)	39.33 (±6.89)	29.48 (±5.17)
Antioxidant activity	FRAP (mg TE, *n* = 7)	1719.40 (±45.92)	1289.58 (±34.44)
	ABTS (mg TE, *n* = 5)	1339.19 (±55.53)	1004.42 (±41.65)
	DPPH radical (mg TE, *n* = 5)	1692.92 (±172.31)	1269.72 (±129.23)

Abbreviations: FRAP (Ferric Reducing Antioxidant Power); ABTS (2,2′-azino-bis (3-ethylbenzo thiazoline-6-sulfonic acid); DPPH (2,2-diphenyl-1-picrylhydrazyl); GAE (Gallic acid equivalents); PAE2 (Procyanidin A2 equivalents); TAE (Tannic acid equivalents); TE (Trolox equivalents). IC_50_: Concentration to give 50% scavenging or inhibition.

## Data Availability

The data presented in this study are available on request from the last author.

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
