# Peer review of "Adansonia digitata* L. (Baobab Fruit) Effect on Postprandial Glycemia in Healthy Adults: A Randomized Controlled Trial"

_nutrients, 2022, doi:10.3390/nu14020398_

Round 1
Reviewer 1 Report
In this study, young healthy volunteers were subjected to an OGTT and some also consumed a drink of baobab fruit extract immediately after the glucose drink. Blood glucose concentrations were lower in the group who received the additional baobab extract (BAE).
The study design is inadequate to form an opinion as to the mechanism of glucose-lowering with the BAE. Adding anything to the glucose drink that might change the rate that it empties from the stomach and/or how it triggers small intestinal feedback mechanisms (e.g. GLP-1 secretion) or insulin secretion will influence the subsequent blood glucose profile (e.g. see Am J Clin Nutr 2007;86(5):1364-8).
The investigators could easily have measured gastric emptying (at least by a stable isotope breath test) and could also have measured insulin and other hormone levels (e.g. GLP-1). There was no adequate control for the BAE drink (e.g. a similar volume drink with similar fibre content). In what way was the study blinded? – subjects surely knew whether or not they were ingesting the BAE drink?
The assumption that the antioxidant properties of the BAE were responsible for glucose-lowering is not well supported by the data. There was a separation of the blood glucose curves already at the 30 min time point – it would be remarkable for this to be attributable to antioxidant activity. The investigators did not measure antioxidant effects in vivo. Antioxidant effects would much more relevant in sustained exposure to the BAE over hours, days or weeks – and this was not evaluated in this acute study.
Author Response
Dear reviewer,
Please see the attachment with a point-by-point response to the comments.
Thank you

Reviewer 2 Report
Rita et al. investigates the effect of an aqueous extract of baobab fruit on postprandial glycaemia on healthy adults, and measures some of its bioactive compounds and antioxidant activity.
The study confirms previous observations about the property of this extract to control glicemia levels. Although the work has a potential interest, the results presented are insufficient and hardly new considering the previous knowledge.
In addition to more deeply analyze the hypoglycemic and antioxidant properties of the extract, as well as its chemical composition, there are some issues regarding the work already carried out that should be improved:
1. Male and female participants in the study were not homogeneously distributed between groups. in fact, the intervention group has a higher female/male ratio. The number of participants must be increased to assure a homogeneous distribution.
2. It could be interesting to differentiate variables in function of sex
3 Lacks some statistical comparison in Table 1
4. The assay was blind for subjects, but not for researchers?. Why?
5. Glycaemic levels before glucose administration must be carefully determined and compared to determine that groups were homogeneous. Lacks a statistical comparison in Table 2 at time 0
6. In addition to mean and SD, individual values of blood glucose levels must be represented in order to known the variability in each group of patients included in the study.
7. Chemical analysis was inadequate. At least one other well-known poliphenolic extract of similar activity should be included, that could be used as a reference. Expressed in this way, the results do not provide relevant information on what consequences they have on the activity of the extract
Author Response

(The authors gave the same response as above.)

Round 2
Reviewer 1 Report
The authors have made some changes to the text, particularly the Discussion, to address the criticisms of the first version, particularly concerning the limitations of the study in determining the mechanism of glucose-lowering. While this has improved the manuscript to some degree, the heavy emphasis on antioxidant properties of BAE which remains in the Discussion (e.g. their potential benefit in preventing vascular complications in diabetes) is disproportionate since this is not really relevant to an acute study in healthy volunteers. A more complete re-write of the Discussion would have been better.
While a paper showing how protein affects the response to oral glucose was suggested in my original review as an example (ref 24), it is not just protein but also any other macronutrient or fibre, and maybe other chemical substances, that could affect gastric emptying and therefore the blood glucose excursion. Rather than simply a comment that the BAE contained little protein, a broader discussion about what determines the blood glucose response to an OGTT is needed.
The concluding sentence of the Abstract simply restates the results and adds nothing.
Substantial revision by a fluent English speaker is also needed, and new typographical errors have been introduced in the revisions. Examples include:
“glycaemia on healthy adults” 9should be “in” – line 29
“sequentially numbers” line 99
“dried environmental local” line 114
“can be stablish” line 339
“as a promisor” line 367
Author Response
Dear reviewer,
please see the attachment.
Thank you
Regards
